# Antioxidant Properties of the Native Khechechuri Pear from Western Georgia

Tamara Gabour Sad [1], Indira Djafaridze [2], Aleko Kalandia [2,*], Maia Vanidze [2], Katarina Smilkov [3] and Claus Jacob [1]

1 Division of Bioorganic Chemistry, School of Pharmacy, Campus B2 1, Saarland University (UdS), D-66123 Saarbruecken, Germany; tamara.sad@outlook.de (T.G.S.); c.jacob@mx.uni-saarland.de (C.J.)
2 Department of Chemistry, Faculty of Natural Sciences and Health Care, Batumi Shota Rustaveli State University (BSU), 35/32 Ninoshvili/Rustaveli str., 6010 Batumi, Georgia; indira.djafaridze@gmail.com (I.D.); vanidzemaia@gmail.com (M.V.)
3 Department of Applied Pharmacy, Division of Pharmacy, Faculty of Medical Sciences, University Goce Delcev (UGD), str. Krste Misirkov No. 10A, 2000 Shtip, North Macedonia; katarina.smilkov@ugd.edu.mk
* Correspondence: aleko.kalandia@gmail.com

**Abstract:** Khechechuri is an endemic species of a pear spread over one region of Western Georgia, called Adjara. Pears are a dietary source of bioactive components such as polyphenols and triterpenic acid. In addition to highlighting its gastronomic value, the aim of the article was to examine and compare phenolic compounds, flavonoids, catechins, phenolic acids, and antioxidant activities in Khechechuri collected from various villages in the Adjara region, namely Adjaristskali, Merisi, Dandalo, Shuakhevi, and Khulo. Five parts of the fruit, the skin, edible pulp, whole pear (skin + pulp), juice, and pomace, were analyzed and the results compared. Our study indicated that the highest total phenolic content was found in the skin of West Georgian pear types (4650 mg/kg.) Moreover, the pomace showed significant amounts of total phenolic content in each of the Khechechuri samples analyzed. Flavonoids were found in each part of the Khechechuri pears, with the notable exception of the fruit juice. A positive correlation between the total phenolic content and the geographical altitude of where the fruits were collected was observed.

**Keywords:** Khechechuri pear; Adjara region; phenolic compounds; catechins; flavonoids; antioxidant activity

## 1. Introduction

Pears are fruits which are widespread across the Globe and have been used in human diet for centuries. Being rich in various macro-, micronutrients, and biologically active compounds, pears represent an important part of the human diet worldwide [1,2]. The Caucasus is a mountain region with extremely diversified natural conditions. A high percentage of endemics is noted in high-mountain and highland xerophytic plant formations. The centers of diversity of the sub-endemic genera are found especially in Georgia, with its Geographical isolation with respect to longitude, latitude and altitude as an important factor in the formation and evolution of the Caucasian flora generated by three-dimensional landscape structure of the mountains [3]. The Khechechuri pear is an endemic type of pear, *Pyrus communis* L., found only in the region of Adjara in Georgia and is very much appreciated for its delicious fruit.

To date, many different studies have examined the composition of this fruit, analyzing the contents of carbohydrates, amino and fatty acids, organic acids and volatile compounds, vitamins and minerals, and bioactive compounds (mainly phenolic content) [1,2,4–7]. The goal of the present work is to assess the antioxidant properties of this endemic type of pear, in terms of examining and comparing the contents of phenolic compounds, flavonoids, catechins, phenolic acids, and antioxidant activities. For this purpose, Khechechuri pears

were collected from five geographically distinct villages in the Adjara region, namely Adjaristskali, Merisi, Dandalo, Shuakhevi and Khulo.

## 2. Materials and Methods

### 2.1. Materials

The pears were collected from the respective villages, measured, and assigned with analysis labels Adjaristskali (A), Merisi (B), Dandalo (C) Shuakhevi (D), and Khulo (E) (Table 1). A total amount of 5 kg of the skin, the edible part (pulp), the whole pear (skin + pulp), juice, and the pomace were homogenized and analyzed. Around 5 g of the mixture of each fruit part was mixed with 150 mL ethanol, and 50 mL of pear extract was used to determine the contents of phenolic compounds, flavonoids, catechins, and phenolic acid in addition to the antioxidant activity. Chemicals used in the methods described here were analytical grade.

**Table 1.** Khechechuri pears collected from different villages with their respective description and characteristics. The fruits were all green in colour with black spots.

| Sample | Appearance | Average Mass (g) | Average Volume (mL) | Size | |
| --- | --- | --- | --- | --- | --- |
| | | | | Average Width (cm) | Average Height (cm) |
| Adjaristskali (A) | | 136.66 | 130.2 | 67.74 | 64.19 |
| Merisi (B) | | 104.43 | 73.3 | 61.87 | 57.51 |
| Dandalo (C) | | 87.77 | 93.3 | 58.78 | 55.79 |
| Shuakhevi (D) | | 150.65 | 67.5 | 68.6 | 63.71 |
| Khulo ((E) | | 123.08 | 145.0 | 65.80 | 60.75 |

### 2.2. Determination of Antioxidant Action

Antioxidant activity was determined by using the DPPH (2,2-diphenyl-1-picrylhydrazil) assay [8,9]. In this context, 1 mL of the prepared sample was added to 3 mL of DPPH extract (0.1 mM DPPH in ethanol), and after 30 min, the change in absorbance at 517 nm

was registered. DPPH and 96% ethanol were used as blanks. For determination of the action of free radical inhibition (DPPH), the following equation was employed

$$In\ \% = \ A_C - \ \frac{A_s}{A_C} \ \times 100 \tag{1}$$

where $A_C$ indicates absorption of the DPPH/alcohol solution, and $A_S$ indicates absorption of the extract [10]. The analyses were performed in triplicate and results expressed as mean value.

### 2.3. Determination of Total Phenolic Content

The total content of phenolic compounds was assessed using the Folin–Ciocalteu method [11,12]. Extraction of the samples was conducted using 80% ethanol; 0.5 or 1.0 mL of the extract was transferred into a 25 mL volumetric flask, and 5.0 mL of distilled water was added along with 1.0 mL of Folin–Ciocalteu reagent. After 8 min at 25 °C, 10.0 mL of 7% $Na_2CO_3$ was added, the flask was then filled to the mark with water and left at room temperature for 2 h. Absorbance was measured at 750 nm. Reagent (1 mL) was used as control. The calculations of the values obtained were made using the calibration curve of gallic acid. For the determination of phenols, the following equation was used:

$$X = (D\ K\ V\ F) \times 1000/m \tag{2}$$

where $X$—amount of phenols (mg/kg); $D$—optical density; $K$—coefficient; $F$—factor of dilution; $V$—volume of extract in mL; $m$—mass of the raw material used for extraction (g). The analyses were performed in triplicates, and results expressed as mean value.

### 2.4. Determination of Catechins

The content of catechin content determined using the Swain and Hill method [13]. A 1 mL aliquot of each sample was pipetted into 3 mL of 1% vanillin reagent (1 g vanillin in 70% sulfuric acid solution in distilled water). Ethanol (1 mL) was used as control. After 15 min, the solution became red, and then after 15 min, the absorption at 750 nm was determined [14]. The analyses were performed in triplicate and results expressed as mean value.

### 2.5. Determination of Total Flavonoid Content

The total flavonoid content (TFC) was determined by the aluminum chloride colorimetric method. The samples (0.5 mL) were mixed with 2 mL of distilled water and 150 μL of 5% $NaNO_2$ solution in distilled water. After 5 min, 150 μL of 10% $AlCl_3$ in distilled water was added in the mixture, and after 6 min, 2 mL 1 mol/L NaOH solution was added. The resulting volume was increased to 5 mL with distilled water. The absorbance was measured at 510 nm and the results expressed in mg/L of catechin [15]. The analyses were performed in triplicate and results expressed as mean value.

### 3. Results and Discussion

Our analyses resulted in variable yet still comparable values of total phenolic content (TPC), total flavonoid content, catechin content, antioxidant activity, and phenolic acid content in the five different parts of Khechechuri pear analyzed, namely fruit, skin, pulp, juice, and pomace. The overall results showed that the highest total phenolic content in all the five different pears examined, was present in their skin. The skin of the five pears was also the richest with the natural antioxidant, catechins, while they scored the lowest in the pulp and juice of the pears. The content of phenolic acids varied according to pear type and was the high in the skin and pomace all the pears. An interesting positive correlation was observed between the total phenolic content and the geographical altitude of where the fruits were collected. The results are accordingly presented in Figures 1–5 and commented upon in each of the following subsections below.

### 3.1. Total Phenolic Content

Figure 1 shows the total phenolic content (TPC) of the Khechechuri pears collected from different areas in the Adjara region. As expected, in each of the collected samples, the highest amount of total phenolic compounds can be found in the skin of the pears, ranging from 4275 to 4644 mg/kg. Indeed, pomace samples showed the second-highest content of phenolic compounds in the samples tested, with concentrations ranging from 3151 to 3717 mg/kg. In contrast, in the pulp, juice and whole fruit, lower percentages of total phenolic compounds were found, 1191–2533, 581–1320, and 1965–2981 mg/kg, respectively.

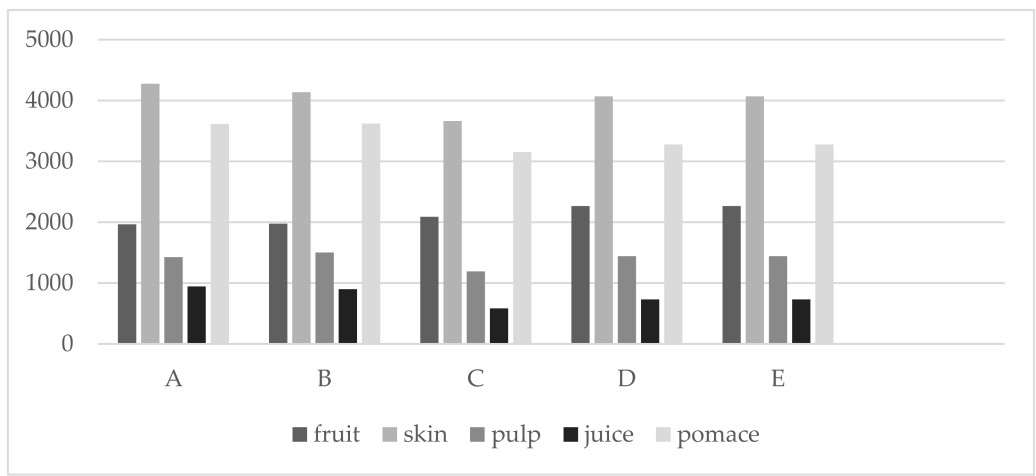

**Figure 1.** Total phenolic content (mg/kg) in different part of the fruits collected from different villages (A) Adjara, (B) Dandalo, (C) Merisi, (D) Shuakehevi and (E) Khulo.

### 3.2. Total Flavonoid Content

Our results show that flavonoids make up to 38–61% of the total phenolic compounds (958–1211 mg/kg), with 29–46% in skin (1336–1734 mg/kg), 39–64% in pulp (657–992 mg/kg), 24–29% in juice (158–319 mg/kg), and 33–68% in pomace (1080–1908 mg/kg) (Figure 2). The content of flavonoids was most pronounced in the skin and pomace in the pears collected from the Adjaristskali region when compared with the other samples tested. These results show that the highest flavonoid content is found in the skin and pomace in each of the pear samples investigated. Unlike these parts, the juice shows a remarkably lower content of flavonoids as depicted in Figure 2.

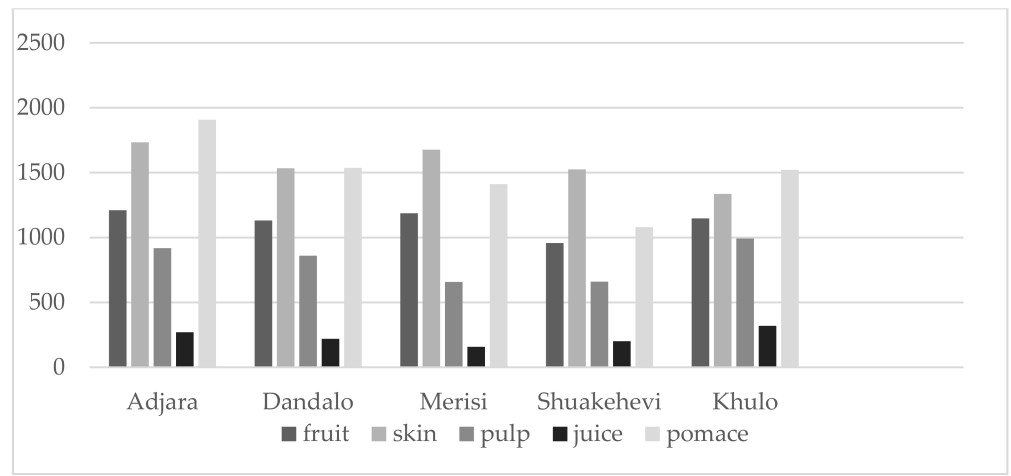

**Figure 2.** Total flavonoid content (mg/kg) of different parts of the fruits collected from different villages. It should be noted that flavonoids account for most of the polyphenolic content and that these flavonoids are notably absent in the juice.

### 3.3. Catechins

The catechins, as a group of flavonoid compounds, were also assessed in these pears. The content of catechins was calculated on the basis of crude mass. As shown in Figure 3, the catechins dominate in the group of flavonoids, and, therefore not surprising, the highest content of catechins was also found in the skin, and here the skin of the pear collected from the Khulo region presented the highest content of catechins, namely 1250 mg/kg. Again, the pomace showed the second-highest content of catechins, and there the pear collected from the Khulo region also provided the highest amount, 576 mg/kg. Catechins were hardly present in the pear juice, ranging from 38 to196 mg/kg in the samples tested.

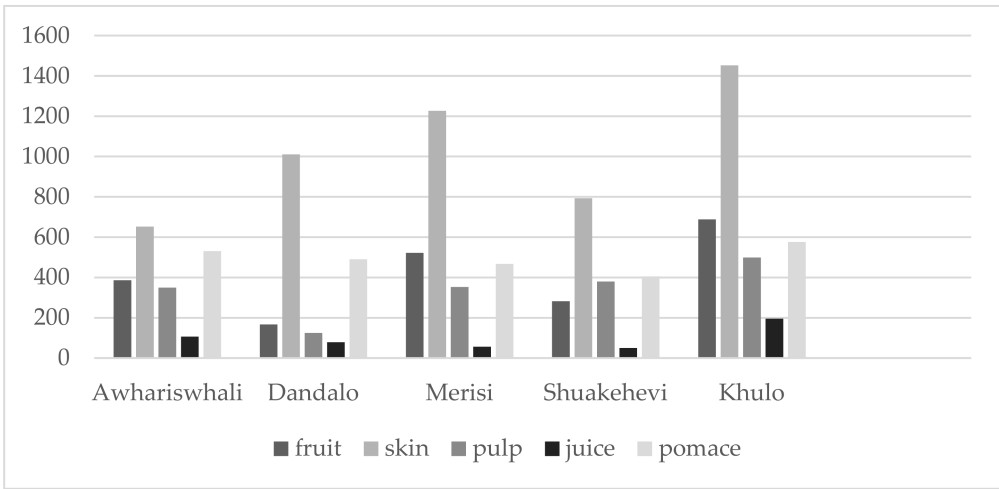

**Figure 3.** Catechin content (mg/kg) of different parts of the fruits collected from five villages, with catechins dominating the flavonoids and present especially in the skin and notably absent in the juice.

In addition, an increase in the height of sea level elevation, i.e. altitude, may increase the concentrations of phenolic compounds, as the projected sea level increases from the village of (a) Adjaristskali, a village in Qeda, at 150 m, (b)Merisi Qeda at 700 m, (c) Dandalo at 520 m, (d) Shuakhevi at 800 m, and (e) Khulo at 950 m. As the pear from Khulo contains most of the active substances and best antioxidant profile, cultivation of the Khechechuri pear at higher altitudes may actually be beneficial. Here it is likely that higher altitudes increase the stress the plants are exposed to and the plants respond to this stress by increased production of antioxidants which may counteract assorted types of plant stresses, including Oxidative Stress.

### 3.4. Antioxidant Activity

Antioxidant activities are expressed in IC 50 values which correspond to the concentration of sample scavenging 50% of DPPH radicals. As shown in Figure 4, antioxidant activity is particularly high in the skin and much lower in the juice, a trend similar to the one observed for the content of flavonoids and catechins, therefore implying that antioxidant activity is mostly the result of the presence of these substances. Notably, the pear collected from Awhariswhali village is more active compared to the others, including the one from Khulo.

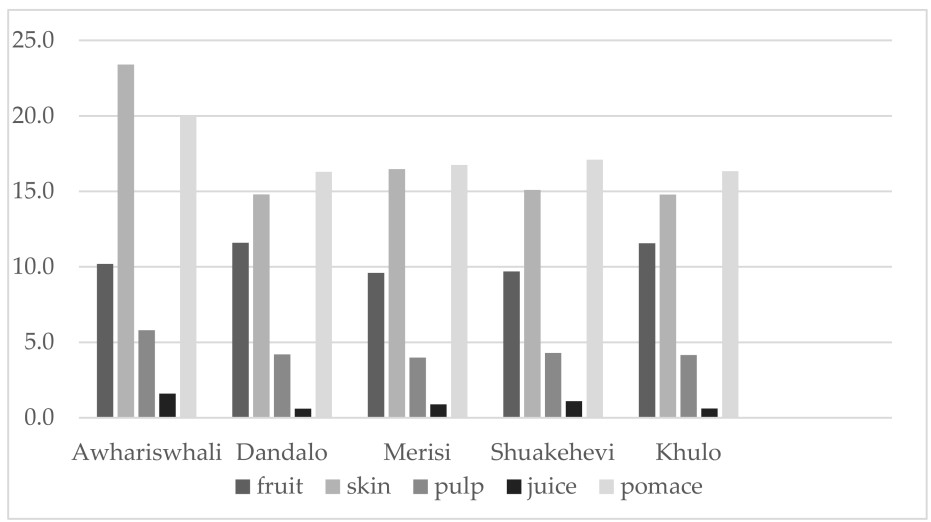

**Figure 4.** Antioxidant activity of the different parts of the fruits from five different villages.

In particular, where the content of phenolic compounds is higher, the antioxidant activity is more likely increased (more than 100 mg sample indicator) (Figure 4).

*3.5. Phenolic Acids*

Skin and pomace include a significant amount of phenolic acids in the samples studied, whereas the pomace was the richest part in the pears collected from Awhariswhali and Dandalo.

The phenolic compounds including phenolic acid were also examined in the samples. An amount of 330–465 mg/kg of phenolic acid was recorded in the fruit.

Compared to other phenolic compounds, a high content of phenolic acid is found in the skin (623–781 mg/kg) and in the pomace (403–703 mg/kg), and only 11–21% (37–78 mg/kg) of the total content of phenolic acid content in the juice (Figure 5).

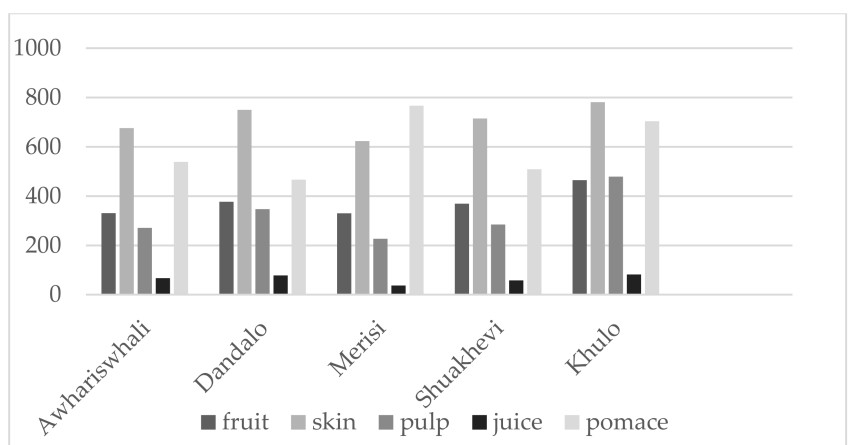

**Figure 5.** Phenolic acid content (mg/kg) of the different parts of the fruits collected from different villages.

**4. Conclusions**

This study has confirmed that the endemic pear Khechechuri is rich in antioxidants, notably flavonoids and here especially catechins. These redox active ingredients provide this delicious pear with considerable antioxidant activity, especially in the skin. Indeed, our results demonstrate that the content of bioactive compounds is higher in the skin and pomace of the Khechechuri pear compared with the pulp and juice, and that this is probably due to the higher water content of the latter which dilutes the concentration of

bioactive compounds and therefore reduces their effectiveness. Other researchers have investigated the total flavonoid and phenolic acid content, in addition to the antioxidant activity of pears found in different regions of the world [16]. Many of these studies have been taking into account the whole fruit, and indeed the studies presented here confirm the antioxidant potential of such pears, especially in the skin. Interestingly, it has also been documented that fruit byproducts, such as peels, seeds, and leaves, contain high levels of various health-enhancing substances, including phenolic compounds [17]. These by-products could be the focus of further research as their utilization may open new possibilities for more sustainable production in the food and pharmaceutical industries. In addition, an increase in the height of sea level elevation may increase the concentrations of phenolic compounds.

Additional experiments are required in the future to investigate the precise mechanism or mechanisms responsible for the pronounced biological activity associated with these Georgian pears

**Author Contributions:** T.G.S. conceptualized, carried out the necessary investigations, visualized and wrote the original and review draft of the manuscript. I.D. carried out the necessary investigations and formal analysis of data and wrote the original draft of the manuscript. A.K. and M.V. supervised this study and wrote the original and review draft of this manuscript. K.S. and C.J. validated and coordinated this study and wrote the original and review draft of the manuscript. All authors have read and agreed to the published version of the manuscript.

**Funding:** This research was funded by Universitaet des Saarlandes, Erasmus code: SAARBRU01". The authors would also like to acknowledge the EU COST Action 16112 "NutRedOx".

**Institutional Review Board Statement:** Not applicable.

**Informed Consent Statement:** Not applicable.

**Data Availability Statement:** The data presented in this study are available on request from the corresponding author.

**Acknowledgments:** The authors would like to acknowledge the effort of Ahmad Yaman Abdin who established the cooperation between the two institutes, UdS and BSU, through the framework of the ERASMUS+ program and the visit awarded to the doctoral student Tamara Gabour Sad from the sending University; University of Saarland, to the receiving University, University of Shota Rustaveli in Batumi, Georgia to conduct this scientific project. The authors express special thanks to Ekaterina Klüh, Ken Rory, Inga Kartsivadze, Tatia Gorgoshadze and many other colleagues of the "Academiacs International" network (www.academiacs.eu) at Saarland University for helpful discussions and inspiration.

**Conflicts of Interest:** The authors declare no conflict of interest.

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
