# Peer review of "Antioxidant Properties of the Native Khechechuri Pear from Western Georgia"

_sci, doi:10.3390/sci3010010_

Round 1

Reviewer 1 Report

In this manuscript, the authors have characterized the phenolic compounds, antioxidant activity, catechines and flavonoids of different parts of endemic pear's species from Caucasus. 

Introduction and material and methods are well redacted and executed in order to get the phenolic information of those samples.

However over-information is noticed in results part of manuscript, exceeding the number of figures. So I recommend to authors reduce or incorporate the figures 9-16 into supplementary data or re-structure those results in tables.

Finally authors concluded well and discussing all results they had in this study.

Author Response

However over-information is noticed in results part of manuscript, exceeding the number of figures. So I recommend to authors reduce or incorporate the figures 9-16 into supplementary data or re-structure those results in tables. Thank you for your comment. After thorough revision of the manuscript, the results section was modified in order to receive more clear and concise context of the manuscript. Therefore, some results that were considered over-informative were removed.

Reviewer 2 Report

The authors presented a research article evaluating different pear types collected in the Republic of Georgia in the Eurasian zone, analysing their shape, weight and aspect. Then the content and identity of various natural compounds have been determined and compared.

Generally speaking, the manuscript is lacking a clear strategy and a logical order. This made it also difficult to write a strictly chronological report, thus I mixed more general points (at the first time of appearance in the manuscript) with more specific points. In my humble opinion the manuscript needs an extensive revision prior to approval.

Title needs to be more informative and precise

Abstract: too much geographic information (this should be included to describe the exact collection points of the pears), furthermore a sound conclusion/scope of the abstract is missing

The general language style is not always scientific and/or easy to understand, thus extensive revisions in this field are necessary.

The graph styles should be unified and more importantly they need a proper statistical analysis (e.g. error bars, uniformity of units, rounding (decimal places), headlines in figures like 3,4 etc are not necessary as they are explained in the relative footnotes).

Regarding the LC-MS, it was stated that some reference compounds/fruits were used, but the relative data are not present in the manuscript? These data must be provided and explained as this point is vital for the quality of the study.

Figure 1 has no citation; can the map be published in the present form (copyright?). It could be furthermore nice to mark the different collection points directly in the map.

Table 1: the form column (circle) makes not much sense, it would be better to replace it with a photo of the relative fruit combining it with figure 2. This brings also one important question: how can you distinguish between these different pears? Can you see a visual difference? Are the mass, size and volume figures a mean value? How many pears have been compared? Are they coming from different species? The precise botanic name (also in Latin), family etc needs to be given for each fruit used in the manuscript.

The overall style (font, size, figures, tables) should be uniform, thus the layout should be carefully checked

Sometimes the authors state “novel pear”, as far as I understood, the pears are existing since long time, but have never been examined/compared scientifically? An explanation/clarification is necessary eventually with references.

The materials and methods section needs to be rechecked carefully. Besides some typos, some words are missing which are crucial for a correct understanding and repetition of the experiments. At 2.4 a reference is mentioned, but note cited properly.

Section 3.4: The first sentence of this paragraph needs to be explained better/inserted better in the context.

Section 3.5: which pears have been studied? It is not very clearly stated. That is “Khechechuri”? In this section the authors mentioned flavones… I assume they mean flavonoids? A part from the database mentioned the authors claim to have used peer-reviewed publications without citing them, thus the citations of the literature used need to be added and discussed also in the introduction.

Compound 1 has not been explained in the text. For all compounds the mass conditions and settings should be reported under the relative images. The axes should be properly labelled and the experiment numbers/peak numbers checked/adjusted.

On page 9 at the bottom suddenly other pears and different varieties compared to the intro are mentioned. It seems that 2 texts have been combined here. This section needs to be properly checked and corrected. It is very important to define the different pears well and in a clear way. Therefore, I suggest to give a number, letter or a sign to each pear type which is then used throughout the manuscript, avoiding misunderstanding.

How do figures 14-16 connect with figure 4? It is also not clear from the text. The precise numbers are already in the table, in the text they are not necessary, it would be more appealing and interesting to the reader to highlt the differences, similarities etc between the different pear types.

The conclusions are rather flaw and do not sum up in a concise way the paper not giving also a proper future perspective. The authors speak about “biological assays”, but they have not been performed? The text does not give a clear answer on that.

Some websites have been cited within the text, they should be implemented in the reference section according to the journal guidelines.

Author Response

Generally speaking, the manuscript is lacking a clear strategy and a logical order. This made it also difficult to write a strictly chronological report, thus I mixed more general points (at the first time of appearance in the manuscript) with more specific points. In my humble opinion the manuscript needs an extensive revision prior to approval. Thank you for your thorough, point-to point review. The manuscript was thoroughly reviewed and the main sections were improved in order to receive a clear and logical order. Title needs to be more informative and precise. The title was changed in order to better represent the topic of the manuscript. Abstract: too much geographic information (this should be included to describe the exact collection points of the pears), furthermore a sound conclusion/scope of the abstract is missing. The abstract was changed. The general language style is not always scientific and/or easy to understand, thus extensive revisions in this field are necessary. The language style was worked on throughout the manuscript. The graph styles should be unified and more importantly they need a proper statistical analysis (e.g. error bars, uniformity of units, rounding (decimal places), headlines in figures like 3,4 etc are not necessary as they are explained in the relative footnotes). The graphs were changed and unified. Regarding the LC-MS, it was stated that some reference compounds/fruits were used, but the relative data are not present in the manuscript? These data must be provided and explained as this point is vital for the quality of the study. Due to some unclear information, after thorough revision of the manuscript parts, this part was removed from the manuscript. Figure 1 has no citation; can the map be published in the present form (copyright?). It could be furthermore nice to mark the different collection points directly in the map. This map was removed. Table 1: the form column (circle) makes not much sense, it would be better to replace it with a photo of the relative fruit combining it with figure 2. This brings also one important question: how can you distinguish between these different pears? Can you see a visual difference? Are the mass, size and volume figures a mean value? How many pears have been compared? Are they coming from different species? The precise botanic name (also in Latin), family etc needs to be given for each fruit used in the manuscript. Table 1 and Fig. 2 were united in one table, in order to receive more clear understanding. Since this is one, endemic species, geographical names of the place of collection was used in order to designate the place the pears were collected. The overall style (font, size, figures, tables) should be uniform, thus the layout should be carefully checked The style was changed and unified. Sometimes the authors state “novel pear”, as far as I understood, the pears are existing since long time, but have never been examined/compared scientifically? An explanation/clarification is necessary eventually with references. Thank you for your comment. Some previous research of the pear species that were published, are mentioned with references in the introduction section. The materials and methods section needs to be rechecked carefully. Besides some typos, some words are missing which are crucial for a correct understanding and repetition of the experiments. At 2.4 a reference is mentioned, but note cited properly. The materials and methods section was reviewed and corrected. Section 3.4: The first sentence of this paragraph needs to be explained better/inserted better in the context. This section has been corrected. Section 3.5: which pears have been studied? It is not very clearly stated. That is “Khechechuri”? In this section the authors mentioned flavones… I assume they mean flavonoids? A part from the database mentioned the authors claim to have used peer-reviewed publications without citing them, thus the citations of the literature used need to be added and discussed also in the introduction. Compound 1 has not been explained in the text. For all compounds the mass conditions and settings should be reported under the relative images. The axes should be properly labelled and the experiment numbers/peak numbers checked/adjusted. This section has been modified. On page 9 at the bottom suddenly other pears and different varieties compared to the intro are mentioned. It seems that 2 texts have been combined here. This section needs to be properly checked and corrected. It is very important to define the different pears well and in a clear way. Therefore, I suggest to give a number, letter or a sign to each pear type which is then used throughout the manuscript, avoiding misunderstanding. How do figures 14-16 connect with figure 4? It is also not clear from the text. The precise numbers are already in the table, in the text they are not necessary, it would be more appealing and interesting to the reader to highlt the differences, similarities etc between the different pear types. This section has been removed. The conclusions are rather flaw and do not sum up in a concise way the paper not giving also a proper future perspective. The authors speak about “biological assays”, but they have not been performed? The text does not give a clear answer on that. The conclusion was modified. Some websites have been cited within the text, they should be implemented in the reference section according to the journal guidelines. The reference section was improved.